# HDAC6 as a Prognostic Factor and Druggable Target in HER2-Positive Breast Cancer

**DOI:** 10.3390/cancers16223752

**Published:** 2024-11-06

**Authors:** Michela Cortesi, Sara Bravaccini, Sara Ravaioli, Elisabetta Petracci, Davide Angeli, Maria Maddalena Tumedei, William Balzi, Francesca Pirini, Michele Zanoni, Paola Possanzini, Andrea Rocca, Michela Palleschi, Paola Ulivi, Giovanni Martinelli, Roberta Maltoni

**Affiliations:** 1IRCCS Istituto Romagnolo per lo Studio dei Tumori (IRST) “Dino Amadori”, 47014 Meldola, Italy; michela.cortesi@irst.emr.it (M.C.); roberta.maltoni@irst.emr.it (R.M.); 2Pathology Unit, Morgagni-Pierantoni Hospital, 47121 Forlì, Italy; 3Department of Medical, Surgical and Health Sciences, University of Trieste, 34127 Trieste, Italy; 4Department of Hematology and Sciences Oncology, Institute of Haematology “L. and A. Seràgnoli”, S. Orsola University Hospital, 40138 Bologna, Italy

**Keywords:** breast cancer, HER2 positive, trastuzumab, adjuvant treatment, resistance, HDAC6

## Abstract

Adjuvant trastuzumab is the standard of care for HER2+ breast cancer (BC) patients. However, >50% of patients become resistant. This study aimed at identifying the molecular factors associated with disease relapse and investigating them as therapeutically exploitable targets. Our findings encourage the exploration of the role of HDAC6 as a prognostic factor and the combinatorial use of HDAC6 selective inhibitors combined with trastuzumab in HER2+ BC, in particular for those patients experiencing drug resistance.

## 1. Introduction

Breast cancer (BC) is the leading cause of malignancy and the second leading cause of cancer-related death in women [1]. Breast tumors with overexpression and or amplification of human epidermal growth factor receptor 2 (HER2) represent 15–20% of all BCs [2]. This BC subtype is biologically and clinically aggressive; it is associated with disease relapse, metastasis, and poor prognosis. Thus, chemotherapy and hormone therapy are insufficient for most patients [3]. HER2-targeting agents such as trastuzumab, lapatinib, pertuzumab, trastuzumab-DM1 (T-DM1), neratinib, tucatinib, and trastuzumab deruxtecan significantly improved patient survival [4,5]. Trastuzumab in the adjuvant setting is administered to adult HER2+ patients—both node-negative and positive—after or in combination with anthracyclines and/or taxanes. Its combination with chemotherapy has been shown to reduce the risk of tumor recurrence by 9.0% when compared with chemotherapy alone [6]. However, a high percentage of HER2+ BC patients exhibit de novo or acquired resistance in less than one year [7]. Molecular mechanisms of resistance to anti-HER2 therapies involve different signaling pathways, summarized in Table 1.

Moreover, downstream signal activation, host immune regulation [14,15], cancer stem cell self-renewal [16], and epigenetic mechanisms have also been associated with trastuzumab resistance [17].

Despite the investigation of multiple biomarkers to predict trastuzumab’s benefit (i.e., growth factor receptor signaling cascades, tumor microenvironment, tumor heterogeneity), up to now, a HER2 status remains the only validated biomarker in clinical practice [18,19]. Thus, to improve the therapeutic effectiveness and patient prognosis, further investigation of the mechanisms underlying trastuzumab resistance in HER2+ BC is necessary.

The main aim of the present study was to identify the molecular factors associated with disease relapse of HER2-positive BC patients treated with adjuvant trastuzumab. To reach the study aim, a case-control study was conducted, and on primary tumors, an extensive gene expression profile specifically designed to evaluate the signatures involved in BC aggressiveness and evolution was performed. The secondary aim was to investigate these molecular factors as druggable targets exploitable in BC.

## 2. Materials and Methods

### 2.1. Patient Selection

Cases and controls were selected among ≥18 years symptomatic patients residing in the Emilia-Romagna region and treated with adjuvant trastuzumab and anthracycline- and/or taxane-based chemotherapy schemes at the IRCCS Istituto Romagnolo per lo Studio dei Tumori (IRST) “Dino Amadori” (Meldola, Italy) between January 2006 and December 2016. All patients had a HER2+ BC diagnosis confirmed by the Anatomic Pathology Department of Morgagni-Pierantoni Hospital of Forlì, Italy. The HER2-positive status was assessed by fluorescence in situ hybridization (FISH) and/or immunohistochemistry (IHC). All patients were HER2 amplified and scored as 2+ or 3+ by IHC. Cases were patients with a disease relapse defined by the presence of ipsilateral, contralateral infiltrating carcinoma, or distant metastases within 5 years from the start of the adjuvant therapy. Controls were randomly chosen among patients without disease relapse within 5 years from the start of the adjuvant therapy and were matched 1:1 with cases by age at the start of adjuvant treatment (±5 years), year of initiation of adjuvant treatment adjuvant treatment (±5 years) and estrogen receptor (ER) status (positive ≥ 1% vs. negative < 1%). All the clinical data were retrieved by reviewing medical charts. The tumor tissue samples and the original hematoxylin–eosin-stained sections marked for the tumor area were obtained from the Anatomic Pathology Department of Morgagni-Pierantoni Hospital of Forlì. IRST and AVR (Area Vasta Romagna) Ethics Committee reviewed and approved the study (approval no. 2157), and written informed consent was obtained from all patients.

### 2.2. Gene Expression Profile

Total RNA was isolated from at least 10 µm thick sections of a formalin-fixed paraffin-embedded primary tumor with the AllPrep DNA/RNA FFPE Kit (Qiagen, Venlo, Limburg, The Netherlands) according to the manufacturer’s instructions and quantified by Nanodrop (ThermoFisher Scientific, Waltham, MA, USA). All FFPE specimens were reviewed by a dedicated pathologist and selected to contain at least 50% of the tumoral cells. Then, RNA was extracted after fine dissection of the tumor component from the FFPE slices. The gene expression profile of the samples was performed with 300 ng of RNA input on the NanoString platform (NanoString Technologies Inc, Seattle, WA, USA) and using the “high sensitivity” setting on the nCounter™PrepStation and 550 field of view (FOV) on the nCounter™Analyzer. The nCounter Breast Cancer 360 Panel (NanoString Technologies) was used according to the manufacturer’s instructions to analyze the expression of 770 genes and important signatures for BC, such as the PAM50 Signature, which classifies breast tumors into four molecular subtypes (Luminal A, Luminal B, HER2-Enriched, and Basal-like) [20]. The panel also includes the tumor inflammation signature (TIS), including 4 areas of immune biology: IFN-ү-responsive genes related to antigen presentation, chemokine expression, cytotoxic activity, and adaptive immune resistance genes. The TIS measures pre-existing, peripherally suppressed adaptive immune responses in the tumor [21]. Data were analyzed using the PAM50 algorithm (NanoString Technologies) and converted into intrinsic subtype calls, risk of recurrence (ROR) scores, and risk categories, as previously described [22]. The nSolver 4.0 analysis software (Nanostring Technologies) was used to perform NanoString advanced analysis, which is able to calculate the pathway scores.

### 2.3. Immunohistochemical Determinations

Four-micron sections of the most representative tumor specimen block for each patient were mounted on positive-charged slides (Bio Optica, Milan, Italy). Immunostaining to detect biomarkers’ expression was performed using the Ventana Benchmark Ultra staining system (Ventana Medical Systems, Tucson, AZ, USA) with the Optiview DAB Detection Kit (Ventana Medical Systems) and ultraView Universal Alkaline Phosphatase Red Detection Kit. Anti-HDAC6 (D2E5) (Cell Signaling) and VENTANA anti-HER2/neu (4B5) (F. Hoffmann-La Roche, Basilea, Switzerland) antibodies were used to stain the cell lines. All the sections were finally counterstained for 16 min with Hematoxylin II (Ventana Medical System) and for 8 min with Bluing Reagent (Ventana Medical Systems). The biomarker expression was quantified as the percentage of immunopositive cells. All samples were evaluated by an expert pathologist.

### 2.4. Cell Lines

SKBR3 and BT474 were purchased from ATCC (ATCC, Manassas, VA, USA). Cell lines were cultured in RPMI (EuroClone S.p.a, Milan, Italy) and DMEM High Glucose (EuroClone S.p.a, Milan, Italy), respectively, supplemented with FBS (10%; EuroClone S.p.a, Milan, Italy) and glutamine (2 mM; EuroClone S.p.a, Milan, Italy). The cells were incubated in a humidified atmosphere containing 5% CO_2_ at 37 °C. Cell lines were authenticated by STR profiling (Cell Line Authentication Service of Eurofins Genomics, Ebersberg, Germany) and routinely tested for mycoplasma. Trastuzumab-resistant subclones were obtained by culturing parental cells with a stepwise increase of the drug over a period of 7–9 months. The final concentration of trastuzumab used was 400 μg/mL.

### 2.5. Doubling Time

For growth analysis, the cells were plated in 6-well plates in triplicate at a concentration of 1 × 10^5^ cells/well. The cells were collected every day and counted for 7 days after plating. The proliferation doubling time was determined by the following formula: log2 (Cv/Cs), where Cv is the number of viable cells at harvest, and Cs is the number of cells seeded. The sum of all previous population doublings determined the cumulative population doubling level at each passage. Cell viability was evaluated through the trypan blue exclusion test, and cell viability always exceeded 95%.

### 2.6. Clonogenic Assay

Following the treatments, 500 cells were seeded in 500 μL of medium in 24-well plates. After 14 days, the resulting colonies were fixed and stained using 0.5% crystal violet in 25% methanol; colonies with more than 50 cells were quantified under an inverted microscope (Olympus IX51 microscope, Olympus Corporation, Tokyo, Japan) by two independent observers. Two series of samples were prepared for each treatment dose. The relative resistance IC_50_ index (RR IC_50_) was calculated as the ratio between the IC_50_ value of the target-therapy-resistant subclone cells and the value of the parental cells.

### 2.7. MTS Assay

Cytotoxicity assays were performed using the CellTiter 96^®^ AQueous One Solution Cell Proliferation Assay (Promega, Milan, Italy). Cells that were seeded onto a 96-well plate were exposed to increasing concentrations of the drugs. The effect of the treatments, alone or in combination, was evaluated after different exposure times, ranging between 3 and 9 days depending on the drug or the drug combination evaluated. Two independent experiments were performed in octuplicate. The optical densities (OD) of the treated and untreated cells were determined at a wavelength of 490 nm using a fluorescence plate reader. Dose–response curves were created using Excel software for Mac v16.89.1, and the IC_50_ values were determined using GraphPad Prism v10.2.

### 2.8. Real-Time RT-qPCR

TRIzol reagent (Life Technologies, Carlsbad, CA, USA) was used, in accordance with the manufacturer’s instructions, for total cellular RNA extraction. The RNA was quantified by the Nanodrop MD-1000 spectrophotometer system. The iScript cDNA Synthesis kit (Bio-Rad Laboratories Hercules, CA, USA) was used to perform reverse transcription reactions; in particular, 400 ng of total RNA was retro-transcribed in 20 μL of nuclease-free water. Real-time PCR was performed by a 7500 Fast Real-Time PCR system (Applied Biosystems, Foster City, CA, USA), and the expressions of the HDAC6 and ERBB2 genes were detected by TaqMan assays. Reactions containing 40 ng of cDNA template, TaqMan Universal PCR Master Mix (2X), and selected TaqMan assays (20X) were carried out in triplicate at a final volume of 20 μL. The samples were maintained at 50 °C for 2 min, then at 95 °C for 10 min, followed by 40 amplification cycles at 95 °C for 15 s, and at 60 °C for 30 s. Gene expression was normalized to the endogenous genes GAPDH, ACTIN-B, and HPRT-1.

### 2.9. HDAC-Glo I/II Assay

The HDAC-Glo I/II assay kit (G6420) was provided by Promega Corporation. Nexturastat A was purchased by Selleck Chemicals LLC (Houston, TX, USA). The WT and HR cell lines were seeded into 96-well white-walled plates at 10,000 cells/well and incubated overnight at 37 °C with 5% CO_2_. In a parallel white-walled 96-well plate, fivefold serial dilutions of trichostatin A (non-selective HDAC inhibitor) and Nexturastat A (HDAC6 selective inhibitor) were prepared. Prior to exposing the cells to the inhibitor, the culture medium was replaced with a serum-free medium. The cells were incubated for 1 h at 37 °C, and then HDAC-Glo I/II reagent was added. After 45 min of incubation, the luminescence signal was read using a BioTek Synergy H1 Multimode Reader (Agilent Technologies, Inc., Santa Clara, CA, USA). HDAC6 activity was indirectly calculated from the assay, subtracting the residual HDAC-Glo I/II activity of the cells after selective inhibition of HDAC6.

### 2.10. Drug Combinations

In the drug combination experiments, cells that were exposed to trastuzumab were treated with serial dilution of Nexturastat A. Sequential treatment schemes and concomitant exposure were evaluated. In sequential experiments, the first drug was washed out before adding the second, and the cytotoxic effect was evaluated immediately after the end of drug exposure. Nexturastat A concentrations were chosen based on previous experiments on HDAC6 (activity measurement and cytotoxicity), while the trastuzumab concentration used corresponded to the plasma peak concentration derived from pharmacokinetics studies (150 μg/mL) [23].

### 2.11. Drug Interaction Analysis

To evaluate the effect of the interaction between drugs, a free web application for interactive analysis and visualization of multi-drug combination response data was used (www.synergyfinder.org) [24]. The cell viability % and recommended default options were used for the four-parameter logistic regression (LL4) curve-fitting algorithm to fit single-drug dose–response curves. The BLISS independent model was used to investigate the degree of combination synergy or antagonism, being the most appropriate when combining multiple concentrations of a drug with a fixed concentration of a second one [25]. Two-dimensional and three-dimensional synergy maps were downloaded from the SynergyFinder website, and the delta scores obtained from the interactive three-dimensional synergy plot were used to describe the interaction between the drug combinations: antagonistic (d score < 10), additive (d score from −10 to 10) or synergistic (d score > 10).

### 2.12. Prediction of NextA Targets Related to BC

Structural information for Nexturastat A (PubChem ID: 71462653) was derived from the NCBI-PubChem database [26]. The structure, in the simplified molecular-input line-entry system (SMILES) format, was submitted to the SwissTargetPrediction database (http://www.swisstargetprediction.ch/) accessed on 19 January 2024. The database allows for the prediction of bioactive compounds based on 2D and 3D structural similarities with known ligands [27]. SwissTargetPrediction is based on the similarity principle, which supposes that two similar molecules are prone to have similar properties. The predicted targets are those having the actives displaying the highest similarity with the query molecule.

BC target genes were derived from the MalaCards database (https://www.malacards.org/) accessed on 19 January 2024, which is an integrated database of the genes associated with human diseases and their annotations [28]. The search term “breast cancer” was used for target research.

### 2.13. Gene Enrichment Analysis

The potential targets of NextA in BC were added as input into the Gene Ontology database enrichment analysis tool (GO) https://geneontology.org/ accessed on 6 November 2023, a comprehensive resource for curated gene sets and a search engine that accumulates biological knowledge [29]. Gene ontology (GO) term enrichment related to the biological processes and molecular function of such targets was conducted. The enrichment results were ranked according to their significance (*p*-value).

Reactome (https://reactome.org/ accessed on 7 November 2023) was used for pathway analysis [30]. Reactome is a freely available, open-source relational database of signaling and metabolic molecules and their relations organized into biological pathways and processes. The most relevant pathways were sorted by *p*-value. Graphs were obtained using SRplot (https://www.bioinformatics.com.cn/srplot accessed on 5 January 2024), a free online science and research plotting service.

### 2.14. Protein–Protein Interaction Network Analysis

The protein–protein interactions (PPI) network map was constructed for the trastuzumab target ERBB2 and NextA targets of BC using the STRING database [31]. The K-means clustering method was applied to identify clusters within the sets of genes. In the maps, proteins are represented as nodes and their interactions as edges. The highest confidence (0.900) was selected for the minimum required interaction score, and disconnected nodes in the network were hidden. The network was subjected to topological clustering analysis using the molecular complex detection (MCODE) Cytoscape plugin [32]. The PPI networks were filtered with a degree cutoff = 2, node score cutoff = 0.2, and K-score = 2 with a maximum depth = 100 to derive the significant modules. The degree and betweenness centrality were calculated to define the role of a node within the network. A Kaplan–Meier survival curve for overall survival (OS) was generated using a Kaplan–Meier plotter (https://kmplot.com/analysis/ accessed on 18 January 2024) using RNA-seq datasets. Only HER2+ BC patients were selected for the analysis.

### 2.15. Statistical Analyses

Data were summarized using mean ± standard deviation (SD), median, first (IQ), and third (IIIQ) quartiles, as appropriate for continuous variables. The categorical variables were reported as a natural frequency and percentage. Correlation among the variables was measured through the Spearman correlation coefficient (the correlation coefficient was r, and the corresponding *p*-values are reported). The association between the clinical covariates and the relapse status was investigated using the Chi-square test or the Fisher’s Exact test, as appropriate, for categorical variables and the Student’s *t*-test or the Wilcoxon–Mann–Whitney test for continuous variables. Gene expression data were normalized using a ratio of the expression value to the geometric mean of all housekeeping genes on the panel. Housekeeper-normalized data were then log2 transformed. To identify the differentially expressed genes between the cases and controls, a moderated *t*-test was applied. Additionally, *p*-values were adjusted using the Benjamini and Yekutieli false discovery rate (FDR) method [33].

Due to the high-dimensional nature of the NanoString data, to identify and combine the genes (and clinical covariates) associated with the risk of relapse, we first performed an unsupervised feature selection using the consensus clustering with the partitioning around medoids algorithm and then applied logistic regression that included the medoids of the obtained clusters and the clinical covariates [34]. The consensus clustering was performed on the 770 genes obtained by the NanoString platform and the optimal number of clusters was determined by inspecting the consensus matrix cumulative distribution. In this study, the medoid is a single representative gene with the highest average pairwise correlation within a cluster. Due to the low frequency of some cells, for the logistic regression analysis, the categories of a few variables were grouped together, such as those for stage and tumor size. In the multivariate analysis, only variables with a univariate *p*-value less than 0.10 were included, except for the lymph node status (due to its strong correlation with the disease stage) and PVI (due to the high presence of missing data) variables. The results are reported in terms of the odds ratios and corresponding 95% confidence intervals. To preliminary explore the predictive accuracy of the multivariable model, the area under the receiver operating characteristics curve (ROC) and the cutoff value on the predicted probabilities were computed using the Youden index criteria.

To explore the prognostic potential of the most promising feature resulting from this study (HDAC6 on independent data), the Breast Invasive Carcinoma cancer dataset of The Cancer Genome Atlas (TCGA) [DOI: 10.1016/j.cell.2018.03.022] was downloaded from cBioPortal (https://www.cbioportal.org accessed on 10 September 2021) [35].

Only HER-2-positive patients who underwent adjuvant therapy were included. The reported outcome was disease-free survival (DFS), defined as the time from diagnosis to disease relapse or the last follow-up time for censored observations. The Kaplan–Meier method was used to estimate the DFS function, the log-rank test for group comparisons, and the Cox proportional hazards model to investigate the magnitude of the association between one or more factors and DFS. The results were reported in terms of hazard ratios (HRs) and corresponding 95% CIs. The median follow-up time was computed using the reverse Kaplan–Meier method. The analyses were carried out with R statistical software version 4.3.1. All of the in vitro experiments were performed at least in duplicate. Statistical analysis was carried out using GRAPH PAD PRISM 9.0 software by applying the Student’s t-test for 2-group comparisons. Differences were considered significant at *p* < 0.05.

## 3. Results

### 3.1. Patients’ Characteristics and Treatment

The analyses were performed on 52 HER2-positive BC patients treated with adjuvant trastuzumab; 26 relapsed (cases) and 26 non-relapsed (controls). The patient clinical characteristics are reported in Table 2. Different adjuvant chemotherapies were combined with trastuzumab in both cases and controls. The combination of anthracyclines and taxanes was the most common regimen. As expected, cases showed a higher stage and size of disease, lymph node involvement, and body mass index (BMI) at the start of adjuvant therapy.

### 3.2. Association Between BC Intrinsic Subtypes and Relapse Status

The analyses on gene expression were performed on 46 subjects because six failed the quality control check. The PAM50 subtype classification defined 16 patients (34.7%) as having HER2-enriched tumors, 17 (37.0%) as luminal B, 9 (19.6%) as luminal A, and 4 (8.7%) as basal BCs (Appendix A). Compared to the control group, the case group accounted for a lower proportion of luminal A subtypes (32% vs. 10%) and a higher proportion of luminal B subtypes (28% vs. 38%). Despite this, the intrinsic molecular subtype classification defined by PAM50, the risk of recurrence score and the risk category were not associated with trastuzumab-treated patients’ outcomes (*p*-value = 0.467, 0.500, and 0.100, respectively). As also reported by other research groups, BC subtypes determined by PAM50 and FISH/IHC were discordant. Gene expression analysis and FISH/IHC result discrepancies could be explained by the different methodologies applied, post-translational regulation, tumor heterogeneity, and sampling techniques [36,37]. However, the subtype classification defined by FISH/IHC was not associated with relapse, with a *p*-value of 0.502. Similarly, the tumor inflammation signature (TIS) score, was not associated with relapse (*p*-value 0.678), and normalized counts for the ERBB2 gene did not significantly differ between relapsed (median value 1.6, IQ–IIIQ: 0.5–3.3) and not relapsed patients (median value 2.5, IQ–IIIQ: 0.4–4.1), with a *p*-value = 0.245.

### 3.3. Association Between Genes, BC360 Signatures and Relapse Status

Through the nCoutner Breast Cancer 360 panel, 48 biological signatures were obtained. From an unsupervised hierarchical clustering, we obtained the relatedness among the signature scores for each sample through an annotated heatmap; three separate clusters, A, B, and C, were identified (Appendix A). Cluster A included patients with higher expression values of inflammatory and immune-related signatures, such as Treg and PD-L2. On the other hand, cluster C included patients with low expression rates of those signatures. Patients in cluster B showed an intermediate profile. AR and PGR signatures trended lower in cases, while B7-H3 and hypoxia trended higher (Appendix A). However, after adjusting for multiple tests, none of the signatures was associated with the relapse status (Appendix A). Appendix A shows the results from the analysis of differential gene expression between the cases and controls for each individual gene. TMPRSS4, GNG4, FGF9, BMP7, FOS, PGR, and TFF3 were differentially expressed in the cases as compared to the controls (Appendix A). However, after adjusting for multiple tests, the differences in gene expression were not significant.

The association between gene expression and clinical features like tumor stage, lymph node status, and tumor dimension were also investigated, and none of the 770 genes were associated with the clinical variables.

Due to the high-dimensional nature of our expression data, the identification and combination of potential prognostic biomarkers were further investigated using consensus clustering on the 770 genes as a dimensionality reduction method, as reported in the Statistical Analysis section. The consensus clustering analysis led to the identification of five gene clusters. The following medoids were obtained: HDAC6 for cluster 1 (n_1_ = 192 genes), WNT4 for cluster 2 (n_2_ = 102 genes), BMPR2 for cluster 3 (n_3_ = 162 genes), PALB2 for cluster 4 (n_4_ = 191 genes), and PARP1 for cluster 5 (n_5_ = 84 genes). STRING pathway correlation analysis of the five medoids resulted in two different clusters (Figure 1A). One related to the DNA repair pathway included HDAC6, PALB2, and PARP1, and one related to the stemness pathway included WNT4 and BMPR2 (Figure 1B).

In the univariate logistic regression analysis, positive lymph node status, higher stage of disease, BMI at the start of adjuvant therapy, and BMPR2 and HDAC6 values were associated with an increased risk of relapse (Table 3). In the multivariate analysis, HDAC6 and disease stage resulted in the main prognostic factors. The AUC that was associated with the predicted probabilities derived from the multivariable model was equal to 0.91 (95% CI: 0.82–1.00), substantially higher than that calculated, including the stage and BMI alone (AUC = 0.69, 95% CI: 0.52–0.85) (Figure 1C).

HDAC6 showed a positive correlation with the apoptosis and TGF beta (signatures (r = 0.51; *p*-value < 0.001; r = 0.37; *p*-value = 0.010, respectively) and a negative correlation with MHC2 and FOXA1 signatures present in the BC360 panel (r = −0.43; *p*-value = 0.003; r = −0.29; *p*-value = 0.052, respectively) (Figure 1D).

### 3.4. Independent Confirmation Analysis of the Potential Prognostic Role of HDAC6

The TCGA data were available for 65 HER2-positive patients treated with adjuvant trastuzumab. However, DFS (7 events total) was available for 60 of the 65 patients. The median DFS was not reached, and the median follow-up time was 40.5 months (95% CI: 32.7–63.3) (Figure 2). Patients’ characteristics are reported in Appendix A. The survival analysis, performed according to the median value of HDAC6 expression, showed that patients with higher HDAC6 values were associated with unfavorable DFS as compared to patients with lower HDAC6 levels (HR = 4.52, 95% CI: 0.83–24.58, *p*-value = 0.081). Adjusting by stage, the dichotomized HDAC6 maintained its association with DFS (HRadj = 5.98, 95% CI: 1.01–35.35, *p*-value = 0.049). The same trend was also found when considering HDAC6 as a continuous variable (HRadj = 1.96, 95% CI: 0.91–4.22, *p*-values = 0.087).

### 3.5. Generation of Trastuzumab-Resistant Subclones

We explored the expression of HDAC6 and ERBB2 genes in a panel of BC cell lines. ERBB2 was expressed in the SKBR3 and BT474 cell lines classified as HER2+ and luminal B, respectively [39], while HDAC6 expression was detected in all cell lines (Figure 3A). Based on these results, we considered the SKBR3 and BT474 cells to perform in the biological evaluation. Both cell lines were sensitive to trastuzumab, as confirmed by the clonogenic assay. The trastuzumab IC_50_ values for both cell lines were lower than 150 μg/mL, corresponding to the plasma peak concentration reported by the pharmacokinetic studies. The BT474 cells were the most sensitive, with an IC_50_ of 38.7 ± 3.06 μg/mL, while the IC_50_ of SKBR3 was 115.2 ± 14.5 μg/mL (Figure 3B). Then, we established trastuzumab-resistant (HR) subclones of the two above-mentioned HER2+ BC cancer cell lines to investigate the therapeutically exploitable role of HDAC6 by using the specific inhibitor NexturastatA. Starting from a concentration of 10 μg/mL, both cell lines were exposed to gradually increasing concentrations of trastuzumab. The cells were considered resistant when able to grow through lifetime exposure using a trastuzumab concentration higher than that corresponding to the plasma peak. The IC_50_ of the SKBR3-HR subclone was 446.9 ± 30.90 μg/mL and 767.5 ± 16.06 μg/mL for the BT47 HR subclone. The relative resistance IC_50_ index (RR IC_50_) of each subclone was calculated: the cell line expressing the highest HDAC6 level showed the highest RR IC_50_ (SKBR3-HR 3.87 vs. BT474-HR 19.8). Once the establishment of resistance was confirmed, SKBR3-HR and BT474-HR were kept at a 400 μg/mL maintenance dose. Simultaneously, the wild-type (WT) parental cell lines were grown without treatment to be used as controls. Both HR subclones showed a growth rate decrease as compared to the parental cell line (Figure 3C). Unless non-significant, we observed the ERBB2 and HDAC6 expression trending up in HR subclones (Figure 3D).

### 3.6. NexturastatA In Vitro Cytotoxic Activity

We then evaluated NextA’s cytotoxic activity on WT and HR cell lines. The cells were treated for 72 h with increasing concentrations of NextA, ranging from 3.7 nM to 100 μM. NextA was effective in both WT and HR cell lines, showing the lowest IC_50_ in the SKBR3 clones (Figure 4A). We then investigated NextA’s inhibition potency on HDAC6 enzymatic activity. To our knowledge, cell-based assays that measure HDAC6 activity in vitro are not available yet; thus, a luminescent assay measuring the whole HDAC classes I and II enzyme activity was performed. Then, the activity of HDAC6 was indirectly evaluated by inhibiting the enzyme with Nexturastat A. The whole basal HDAC I and II activity was similar in the SKBR3 and SKBR3-HR cells, while in BT474-HR, the enzymatic activity was lower compared to the parental cell line (Figure 4B). Figure 4C shows that NextA effectively inhibited HDAC6 activity in both WT cell lines and retained its inhibitory activity in the HR subclones. Low concentrations of the molecule selectively inhibited HDAC6 without inducing a cytotoxic effect, while increasing drug concentration led to cell death.

### 3.7. Effect of the Selective Inhibition of HDAC6 Combined with Her2 Blockade on BC Cells

On the basis of these results, we investigated the combination of trastuzumab with sub-cytotoxic concentrations of NextA. The two drugs showed a synergistic effect when administered simultaneously in both parental cell lines. In HR subclones, the combination was synergistic in BT474 and additive in SKBR3. Notably, in both HR subclones, the efficacy of the combination increased proportionally with the NextA concentration increase (Figure 5). When trastuzumab and NextA were administered in sequence—trastuzumab followed by NextA and vice versa—we observed different effects: antagonism in both SKBR3 WT and HR clones; and additivity or antagonism in BT474, depending on the clone (Appendix A).

### 3.8. Prediction of NexturastatA Targets Involved in BC

To further investigate the synergism observed in vitro, we used a network pharmacology approach. An outline of the tools exploited for the network pharmacology study is reported in Figure 6A. The biological targets of Nexturastat A were predicted using the SwissTargetPrediction database, and 100 targets were identified [40]. Meanwhile, we obtained from the MalaCards database 1055 BC-associated genes. The intersection of Nexturastat A putative targets against BC-associated genes revealed an overlap of 38 genes among the 100 targets (Figure 6B). The complete list of NextA targets and BC-associated genes is reported in the Appendix A.

### 3.9. Gene Enrichment Analysis of Target Genes

We performed an enrichment analysis of the top GO terms to identify the targets involved in the biological processes and to know their molecular functions [41]. The results revealed an association of the NextA targets of BC with protein phosphorylation, regulation of phosphate metabolic processes, and regulation of cell motility and migration.

Moreover, they were found to be associated, among others, with molecular functions like catalytic activity and kinase activity. The full list of enriched biological processes and molecular functions is reported in the Appendix A.

The same gene list was also submitted to Reactome [42]. The most relevant pathways enriched within the submitted data were signaling by receptor tyrosine kinases, diseases of signal transduction by growth factor receptors and second messengers, and NOTCH-related pathways. The most relevant pathways were sorted by *p*-value (Figure 6C). The full list of pathways is reported in the Appendix A.

### 3.10. Protein–Protein Interaction Networks of Trastuzumab and NextA Targets of BC

A protein–protein interaction (PPI) network map was constructed for BC targets of both trastuzumab and NextA to provide a rationale for the observed drug synergism (Figure 6D). The number of connections of a node to other nodes is described by the degree. The higher the degree, the greater the number of connections with other nodes of the network. On the other hand, the betweenness centrality indicates the importance of a node connecting other nodes within the network. Targets with a larger degree and betweenness centrality represent the essential nodes of the network. The average degree observed in the PPI network map was 2.7, and the betweenness centrality was 0.101.

Importantly, only five genes were identified above the averages: HSP90AA1, NCOR, SRC, HDAC2, and CDK4. Such targets represent the essential node of the network. HSP90AA1 showed the highest betweenness centrality and the highest degree (Figure 6E). This data indicates a potential role of these genes in the synergistic activity of trastuzumab combined with NextA in BC.

### 3.11. Survival Analysis of Putative Synergistic Targets

To evaluate if the expressions of HSP90AA1, NCOR, SRC, HDAC2, and CDK4 were associated with overall survival (OS) among HER2+ BC patients, we used the Kaplan–Meier Plotter (https://kmplot.com accessed on 18 January 2024) to perform survival analysis.

Interestingly, only HSP90AA1 was associated with OS (*p*-value < 0.05). Moreover, HER2 + BC patients with higher HSP90AA1 values demonstrated a significantly unfavorable OS than patients with lower values (HR = 2.37; 95%CI = 1.27–4.45; *p*-value 0.006) (Figure 7).

## 4. Discussion

The response rate, disease progression, and OS are closely associated with BC heterogeneity. Many patients with advanced HER2-positive breast cancer experience either de novo or acquired resistance to anti-HER2 therapy within less than a year, although the mechanisms behind this resistance are not yet fully understood.

Thus, to improve the therapeutic effectiveness and patient prognosis, further investigation into the mechanisms underlying trastuzumab resistance and new therapeutic strategies in HER2+ BC is necessary.

In addition to anti-HER2 agents, several emerging alternative therapies and novel approaches are being researched for HER2-positive breast cancer. Recently, trastuzumab deruxtecan (T-DXd) was approved by the FDA for adult patients with unresectable or metastatic HER2-positive solid tumors who have received prior systemic treatment and, without satisfactory alternative treatment options, this shows superiority over T-DM1 [43,44]. For patients with brain progression in the second or third line, the combination of tucatinib, trastuzumab, and capecitabine has proven effective in the phase III HER2Climb-01 trial [45]. HER3, a member of the human epidermal receptor family, is emerging as an actionable target in BC. Both monoclonal and bispecific antibodies targeting HER3, including zenocutuzumab (MCLA-128) and patritumab deruxtecan (U3–1402; HER3-DXd), are under development and have achieved impressive response rates in phase I/II clinical trials [46]. Moreover, the combination of anti-HER2 agents with immunotherapy (i.e., pembrolizumab plus trastuzumab, trastuzumab emtansine plus atezolizumab and avelumab), despite disappointing clinical trial outcomes, suggests a synergistic effect in HER2-positive advanced breast cancer [47,48]. Therapeutic vaccines are another promising strategy, although these are still at a relatively early stage of development [49].

Despite the limited number of cases analyzed, our results are consistent with the data present in the literature about trastuzumab resistance in the adjuvant setting, where no relationship was observed between the expression levels, copy number of HER2, and patient’s outcome, neither for the PI3KCA mutations or PTEN loss [50]. In addition, similar to the results of the NSABP-B31 trial in which there was no difference in the degree of benefit from adjuvant trastuzumab and BC intrinsic subtypes identified by PAM50, we did not find any association between the PAM50 signature and trastuzumab benefit [51].

In the present study, histone deacetylase 6 (HDAC6) was identified as an independent prognostic marker in HER2+ BC patients treated with adjuvant trastuzumab. Patients with high HDAC6 levels showed a worse prognosis than those with low HDAC6 levels. HDAC6 is a distinctive member of the type II HDAC family, primarily found in the cytoplasm [52]. It targets several non-histone substrates, including α-tubulin, cortactin, and heat shock protein 90 (HSP90), playing a role in regulating tumor cell proliferation, metastasis, invasion, and mitosis [53]. Additionally, HDAC6 has been shown to influence the expression of specific tumor-associated antigens, MHC class I proteins, co-stimulatory molecules, and cytokine production and appears to be a key regulator of the STAT3 pathway [54]. Recent studies have indicated that the histone deacetylase inhibitor trichostatin A works synergistically with lapatinib, a HER2 tyrosine kinase inhibitor, to inhibit breast cancer in vitro and in rodent models. These findings point to a potential new mechanism of HER2-driven carcinogenesis, suggesting that combining HER2 and HDAC targeting could represent a valuable therapeutic strategy [55]. Given its unique structure and functions, HDAC6 is being considered a promising cancer therapeutic target, with four HDAC inhibitors—vorinostat, romidepsin, belinostat, and panobinostat—approved for various cancers. However, there is limited research on the effects and mechanisms of selective HDAC6 inhibitors (HDAC6i), specifically in breast cancer [56].

The overexpression of HDAC6 has been reported in both hematological and solid tumors [57,58]. However, alterations of HDAC6 in tumors are not only related to its expression level but also to its activity [59,60]. To explore the potential role of HDAC6 as a therapeutically exploitable target in BC, a trastuzumab-resistant subclone of two HER2+ BC cell lines—one expressing HDAC6 at low levels (SKBR3) and one at high levels (BT474)—was established. We observed slight differences in the HDAC6 mRNA and protein expression levels among parental and resistant subclones; however, differences were observed in the enzyme activity. Exposing cells to Nexturastat A, a selective HDAC6 inhibitor, we observed that low concentrations of the molecule selectively inhibited HDAC6 without inducing a cytotoxic effect while increasing the concentrations (>of 1 μM for SKBR3 and >of 12.5 μM for BT474) induced dose-dependent cell death. Cell death, beyond HDAC6 inhibition, could also be blamed on a non-specific effect of the molecule on other HDAC isoforms, magnified by the increased drug concentration [61]. The loss of selectivity may lead to toxicity issues, which limit the clinical application of these inhibitors, especially of pan-HDAC inhibitors [62]. The main pan-HDACi toxicities reported are nausea/vomiting, fatigue, and a transient decrease in platelet and white blood cell counts. Other reported adverse effects are cardiotoxicity, liver toxicities, electrolyte imbalances, and neurological events. Of note, these adverse events have been reported with the use of pan-HDACi as single agents and are likely to worsen in a drug combination regimen [63].

With this concept in mind, we investigated the combination of trastuzumab with sub-cytotoxic concentrations of NextA. Different schemes of treatment were evaluated. In general, synergism appeared to be highly dependent on the administration sequence and on NextA concentrations. The combinations were more effective in cells highly expressing HDAC6 (BT474 and BT474-HR). The simultaneous administration of the drugs resulted in being more effective than other sequences. However, in parental cell lines, synergism was observed independently from the trastuzumab/NextA ratio, while in HR cells, the synergistic effect increased proportionally with the NextA concentration increase. Treatment sequences were ineffective, even showing an antagonistic effect in low-expressing HDAC6 cells, while in highly HDAC6-expressing cells, they were effective, at least in the HR subclone. Our data are consistent with the results from other research groups, which show that HDAC6 inhibitors could be safely combined with other anti-tumor drugs and reporting the HDAC6 score as a potential predictive biomarker [57]. We are aware that, up to now, there are no established cutoff values to establish HDAC6 positivity. The search for a good anti-HDAC6 antibody and unique criteria to establish HDAC6 positivity remain an open question and an unmet need. Potential strategies for addressing these challenges may encompass studies involving large numbers of patients where data on HDAC6 expression can be collected continuously, allowing for more accurate and detailed assessments of their associations with survival. Additionally, for the selection of the best HDAC6 IHC cutoff value, future studies will be necessary to define the percentage of immunopositive tumor cells useful to distinguish patients with different disease behaviors. Moreover, the evaluation of the H-score, defined as the staining intensity multiplied by the percentage of immunopositive tumor cells, could also be more informative in terms of prognosis prediction with respect to the percentage of immunopositive tumor cells alone.

The mechanisms of action of HDAC inhibitors and the biological effects of HDAC inhibition are far from being completely understood [64]. Inhibiting individual HDAC isoforms is emerging as a well-tolerated anticancer strategy compared with pan-HDAC inhibitors [65]. However, only limited therapeutic success has been achieved with HDAC inhibitors as single agents, while the combinatorial use of these molecules returned encouraging results that overcame undesired effects [66]. To elucidate the possible mechanisms behind the anti-tumor activity observed by combining trastuzumab and NextA, a network pharmacology approach was applied. First, BC-specific targets of NextA were identified. A total of 38 genes were associated, among other pathways, with pathways associated with BC pathogenesis [67]. Then, a PPI network map was constructed according to STRING’s minimum required interaction score. Notably, 22/38 NextA target proteins and HER2, the trastuzumab target protein, clustered in three groups connected by a central node represented by the protein HSP90AA1. It is well known that HDAC6 deacetylates HSP90, resulting in the protection of oncoproteins from proteasomal degradation [68]. At the same time, HDAC6 is also an HSP90 client protein whose stability is modulated by HSP90 itself [69]. Moreover, HSP90 also directly modulates HER2 kinase activity, which is supposed to play a major role in trastuzumab resistance [70]. Based on the known molecular functions of HDAC6, HER2, and HSP90, we can speculate that the synergism observed may be exploited by HDAC6 modulation of the acetylation status of HSP90 clients, including HER2, leading to the destabilization of the latter and enhancing the efficacy of anti-HER2 therapies; and HDAC6’s influence on HSP90’s chaperone function, affecting its ability to refold misfolded proteins and promoting degradation of HER2.

Our study has some limitations that are worth discussing. Even if we obtained encouraging results, an independent validation in a larger study to strengthen the findings is needed. Then, pathway analysis of HDAC6 and its synergistic link with anti-HER2 therapies has been mainly investigated by computation tools and needs to be experimentally validated. Moreover, the use of different anti-HER2 drugs with different HDAC6 inhibitors should be investigated to identify the best combination strategies.

Notwithstanding the need for result validation, these pieces of evidence shed light on the therapeutic exploitation of HER2 plus HDAC6 inhibition, probably involving HSP90 as a key node of the combined inhibitors’ synergistic activity.

## 5. Conclusions

This study highlighted the possible role of HDAC6 as a prognostic marker in HER2+ BC patients treated with adjuvant trastuzumab. Patients treated with trastuzumab with high HDAC6 levels showed a worse prognosis than those with low HDAC6 levels. Nevertheless, evidence was provided about the therapeutic benefit of the combinatorial use of trastuzumab and the HDAC6 inhibitor NextA in HER2+ BC patients, including those undergoing trastuzumab resistance. HSP90AA1 was also supposed to play a role in the observed synergism. However, for a translation into potential clinical applications, those findings must be validated. The validation of those findings would pave the way for an investigation of effective combinatorial targeted therapies with the goal of improving patients’ survival, in particular for those experiencing trastuzumab resistance (Figure 8).

## Figures and Tables

**Figure 1 cancers-16-03752-f001:**
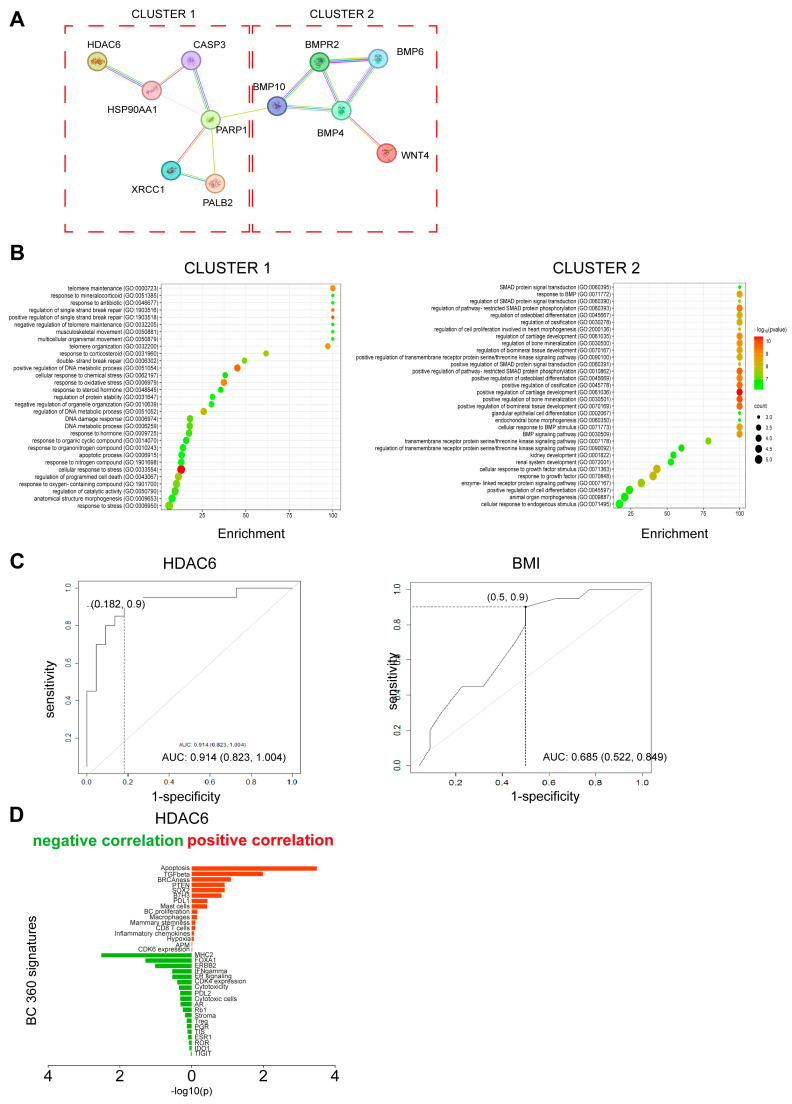
Identification of prognostic markers. (**A**) Protein–protein interaction (PPI) network of the five medoids based on STRING database. The network nodes are proteins. The edges represent the predicted and known protein–protein associations. Different colored edges represent different associations. Light blue: experimental evidence from the database; purple: experimentally determined; yellow: text mining evidence; indigo: protein homology. (**B**) Enrichment bubble plot describing the pathways enriched within clusters 1 and 2. Cluster 1 is composed of 6 genes (HDAC6, PALB2, PARP1, XRCC1, CASP3, and HSP90AA1). Cluster 2 is composed of 5 genes (BMP10, BMPR2, BMP4, BMP6, and WNT4). Information on pathway enrichment was retrieved using GO enrichment analysis tools (https://geneontology.org accessed on 6 November 2023). The color scale indicates different thresholds of the *p*-value, and the size of the dot indicates the number of genes corresponding to each pathway. Bubble plots were obtained using the online free tool SRplot [38]. (**C**) ROC curve for prediction of disease relapse based on HDAC6 gene expression and on BMI. Curve from the multivariable logistic model. Sensitivity and 1-specificity values were also reported in correspondence with the cutoff obtained by the Youden index. (**D**) Diagram showing the correlations between HDAC6 and BC360 signatures. Signatures positively correlated are shown in red, and signatures negatively regulated are shown in green. The asterisks indicate the most significant correlation based on *p*-value and R score (−0.4 ≥ R score ≥ 0.4).

**Figure 2 cancers-16-03752-f002:**
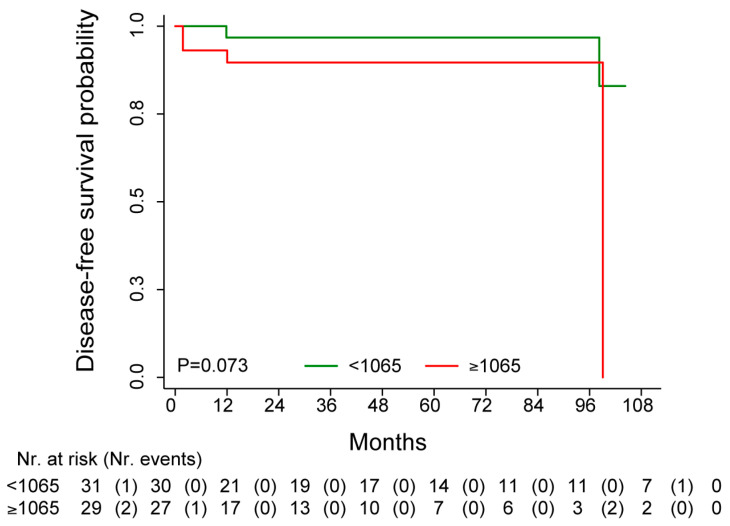
Kaplan–Meier curve for DFS in patients with HER2 + BC treated with adjuvant trastuzumab stratified by median HDAC6 expression value.

**Figure 3 cancers-16-03752-f003:**
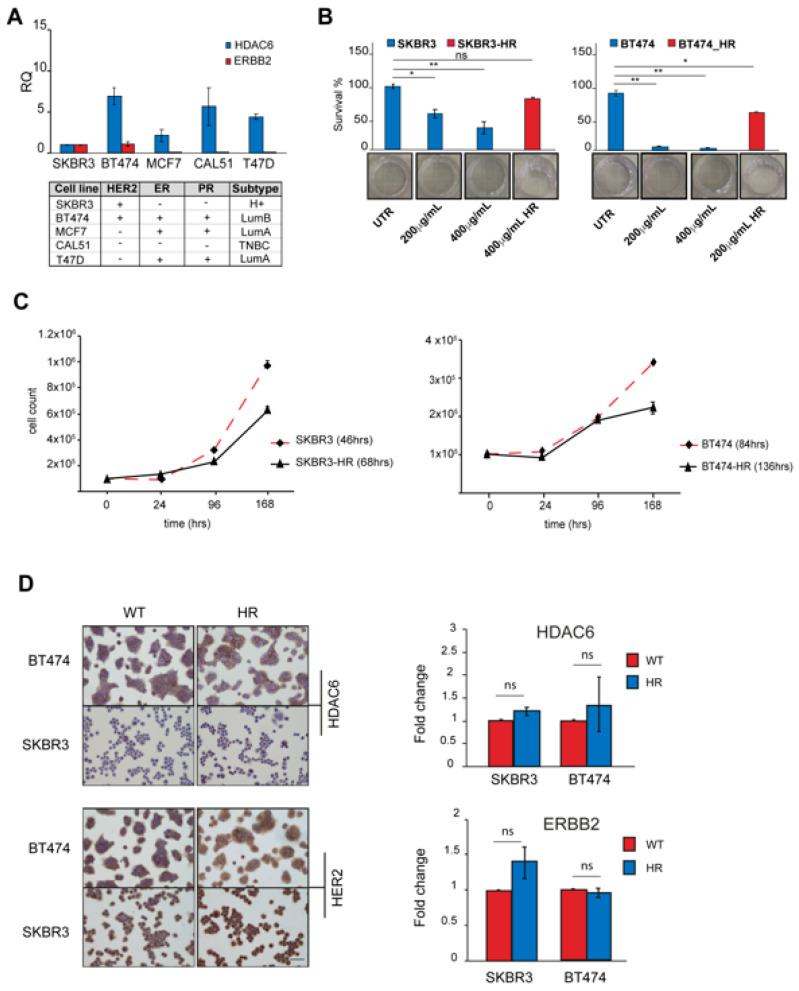
Generation of trastuzumab-resistant subclones. (**A**) HDAC6 and ERBB2 relative gene expression analysis performed by quantitative real-time RT-PCR. Expression levels in both genes were normalized to GAPDH, HPRT-1, and ACTB. The bars show RQ ± standard deviation (SD). * *p* < 0.05; ** *p* < 0.01. (**B**) Clonogenic assay. BC cell lines, both WT and HR, were treated with trastuzumab at concentrations of 200 mg/mL and 400 mg/mL for two weeks. Survival fractions were obtained by counting colonies with more than 50 cells by inverted microscope (n = 12 for each time point, mean ± SD; ns: not significant). (**C**) Cell doubling time (DT). BC cell lines (WT and HR) were counted every day until confluence. Population doubling time was calculated using the formula log2 (Cv/Cs). n = 3 for time point mean ± SD). (**D**) Left panel: representative images of IHC staining for HDAC6 and HER2 of WT and HR cell lines. Images magnification 20×. Right panel: relative gene expression analysis performed by quantitative real-time RT-PCR. Expression levels of HDAC6 and ERBB2 were normalized to GAPDH, HPRT-1, and ACTB. The bars show fold change ± standard deviation (SD) ns: not significant.

**Figure 4 cancers-16-03752-f004:**
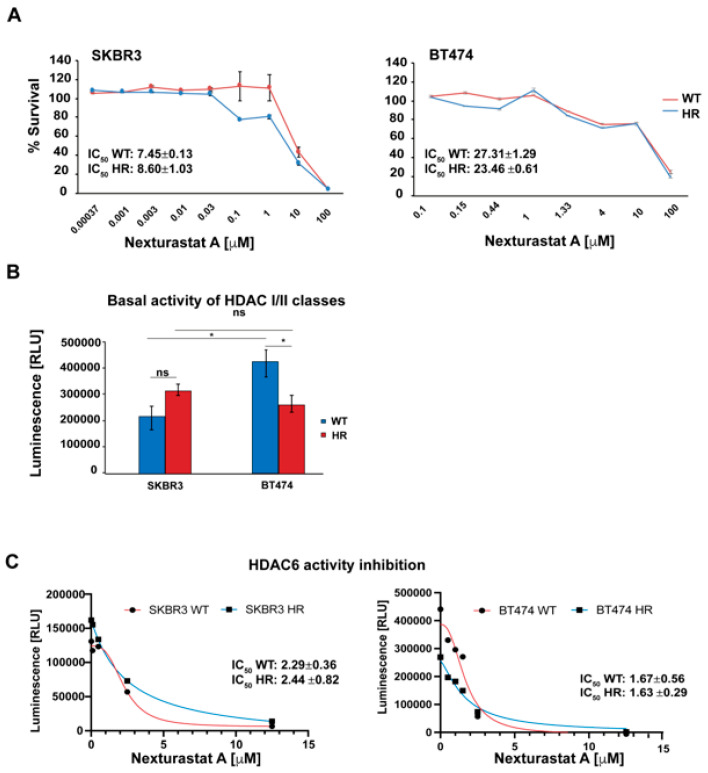
HDAC6 activity in BC cells. (**A**) NextA cytotoxic activity. BC cells were treated with NextA at concentrations ranging from 0.37 nM to 100 μM for 72 h. n = 3, mean ± SD. Cell viability was measured using MTS CellTiter96^®^ Aqueous One Solution cell proliferation assay. (**B**) Whole HDAC classes I/II basal activity in both WT and HR BT474 and SKBR3 cell lines, * *p* < 0.05, ns: not significant; (**C**) WT and HR cells were treated with Nexturastat A at concentrations ranging between 0.008 μM and 12.5 μM. HDAC class I/II activity was measured using HDAC-GloTM class I/II assay.

**Figure 5 cancers-16-03752-f005:**
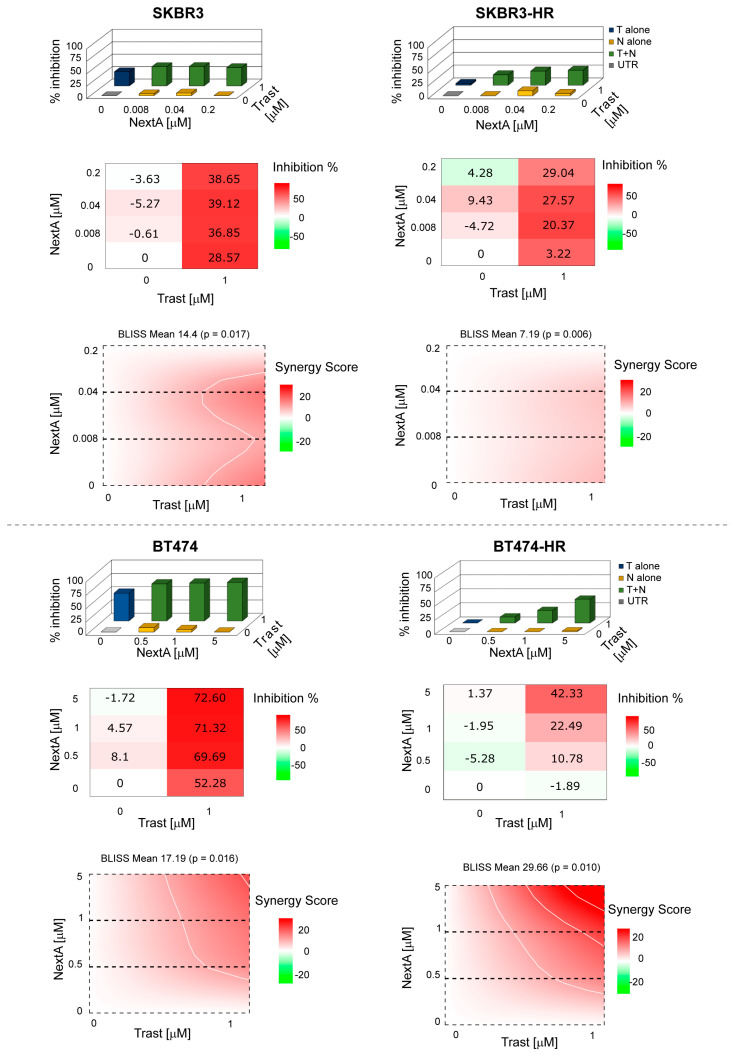
Drug combination evaluation. BC cells were treated with trastuzumab and NextA both alone and in combination. MTS CellTiter96^®^ Aqueous One Solution cell proliferation assay was used to determine cell viability. The BLISS synergy score was evaluated using the SynergyFinder.org web application. Score < −10 antagonism, between −10 and 10 additivity, >10 synergism.

**Figure 6 cancers-16-03752-f006:**
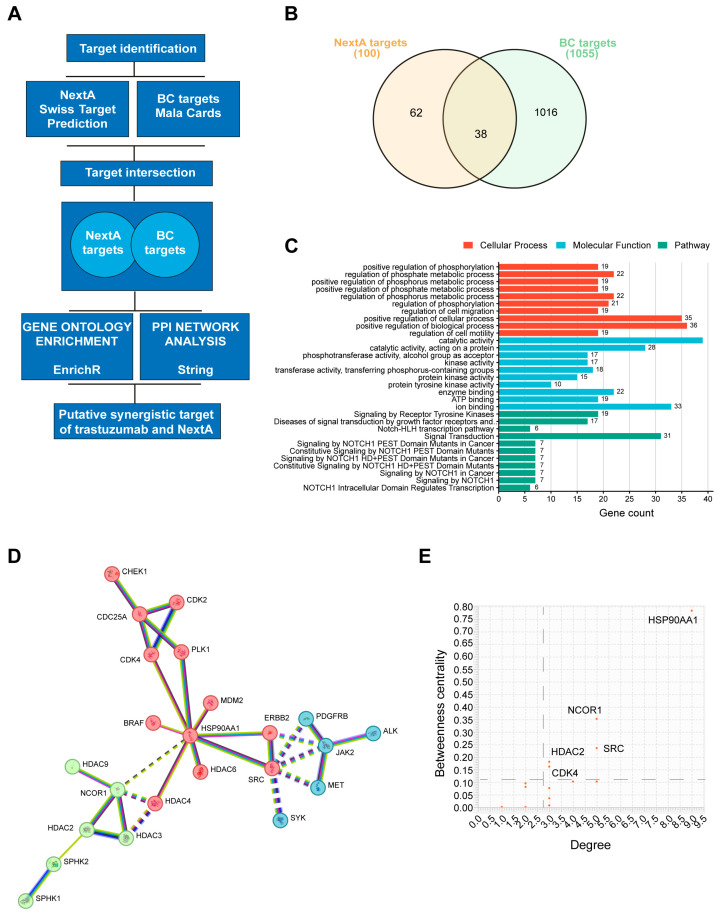
Network pharmacology. (**A**) Outline of the tools used for the study. (**B**) Venn diagram represents the intersection of BC-associated genes and NextA targets. (**C**) Bar plot showing most significant cellular processes, molecular functions, and pathways enriched with NextA and BC-associated common genes. (**D**) PPI network map of trastuzumab and NextA targets against BC targets. The unsupervised clustering method, k-means clustering, was applied to identify the clusters. (**E**) Graph showing the most relevant targets of the PPI network identified by degree and betweenness centrality.

**Figure 7 cancers-16-03752-f007:**
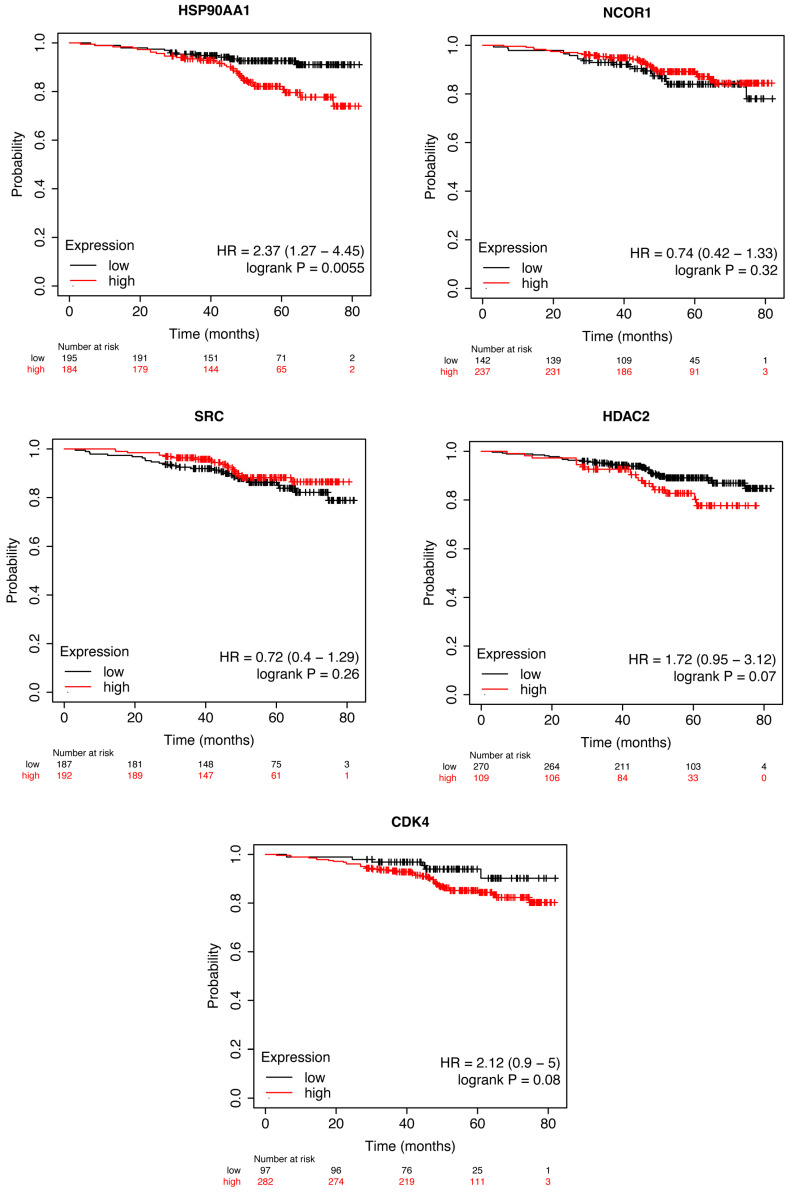
Kaplan–Meier curves for OS in patients with HER2 + BC stratified HSP90AA1, NCOR1, SRC, HDAC2, and CDK4 expression values.

**Figure 8 cancers-16-03752-f008:**
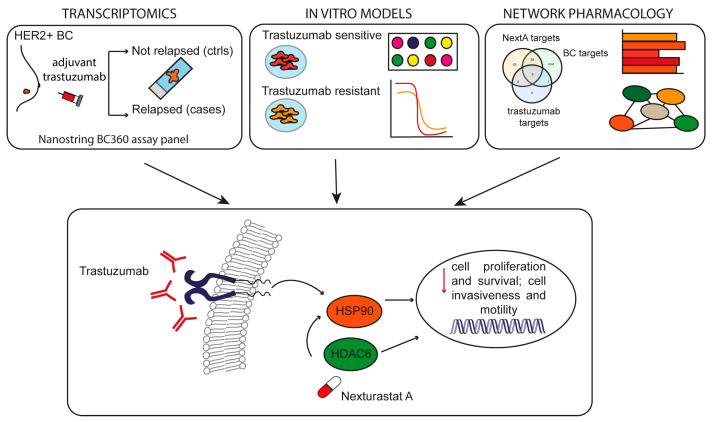
Graphical abstract. Using transcriptomics, in vitro models, and network pharmacology, we found that trastuzumab synergizes with the HDAC6-specific inhibitor NexturastatA, probably through the modulation of HSP90, leading to cell death and inhibition of cell proliferation of HER2+ BC cells that are both sensitive and resistant to trastuzumab.

**Table 1 cancers-16-03752-t001:** Molecular mechanisms of resistance to anti-HER2 therapies.

Mechanism	Description	Ref
Signaling from other HER receptors	Activation of HER3 or EGFR can bypass HER2 blockade, sustaining growth.	[8]
Activation of PI3K/AKT/mTOR pathways	Promotes cell survival and proliferation, contributing to resistance.	[9]
C-MET overexpression	Activates alternative pathways, enhancing tumor survival.	[10]
Loss of PTEN	Leads to increased PI3K signaling, promoting tumor growth.	[11]
Upregulation of Src activity	Enhances tumor cell proliferation and survival through various pathways.	[12]
MUC4 expression	Inhibits binding of trastuzumab to HER2, reducing its efficacy.	[13]
Expression of p95 HER2 isoform	Lacks trastuzumab binding site, allowing continued signaling.	[14]

**Table 2 cancers-16-03752-t002:** Patients’ characteristics (n = 52).

Variable	Cases (n = 26)		Controls (n = 26)	
	n	(%)	n	(%)
**Age at start of adjuvant therapy (years)**				
Mean ± SD	53.1 ± 12.6		53.7 ± 11.4	
**Menopausal Status**				
Pre-menopause	10	(38.5)	11	(42.3)
Post-menopause	16	(61.5)	15	(57.7)
**Stage of Disease**				
I	2	(8.0)	7	(30.4) **
II	7	(28.0)	13	(56.5)
III	16	(64.0)	3	(13.0)
Unknown	1		3	
**Lymph Node Status**				
pN negative	4	(16.7)	14	(53.9) **
pN positive	20	(83.3)	12	(46.2)
Unknown	2		0	
**Tumor Size (at Diagnosis)**				
pT1	7	(26.9)	21	(80.8) **
pT2	8	(30.8)	5	(19.2)
pT3	6	(23.1)	0	(0.0)
pT4	5	(19.2)	0	(0.0)
**Tumor Histology**				
Ductal carcinoma	22	(84.6)	23	(88.5)
Lobular carcinoma	1	(3.9)	2	(7.7)
Other	3	(11.5)	1	(3.9)
**Histologic Grade**				
1	1	(4.2)	1	(4.0)
2	8	(33.3)	10	(40.0)
3	15	(62.5)	14	(56.0)
Unknown	2		1	
**Ki-67**				
<20%	8	(30.8)	9	(34.6)
≥20%	18	(69.2)	17	(65.4)
**ER**				
Negative	7	(26.9)	7	(26.9)
Positive	19	(73.1)	19	(73.1)
**PgR**				
Negative	11	(44.0)	11	(42.3)
Positive	14	(56.0)	15	(57.7)
Unknown	1		0	
**VI**				
No	3	(37.5)	10	(66.7)
Yes	5	(62.5)	5	(33.3)
Unknown	18		11	
**Adjuvant Chemotherapy in combination with Trastuzumab**				
Anthracyclines and taxanes	7	(26.9)	17	(65.4) **
Taxanes only	4	(15.4)	2	(7.7)
Anthracyclines only	3	(11.5)	7	(26.9)
Other (endocrine therapy, CMF, vinorelbine)	12	(46.2)	0	(0.0)
**BMI at start of Adjuvant Chemotherapy**				
Mean ± SD	25.9 ± 3.2		23.6 ± 4.4 *	

** *p*-value < 0.01; * *p*-value < 0.05; SD: standard deviation; ER: estrogen receptor; PgR: progesterone receptor; VI: vascular invasion; BMI: body mass index; CMF: Cyclophosphamide Methotrexate Fluorouracil.

**Table 3 cancers-16-03752-t003:** Results from univariate and multivariate logistic regression analyses.

	Univariate			Multivariate
Variable	OR	(95% CI)	OR	(95% CI)
**Age at start of adjuvant therapy (yrs)**	0.99	(0.94–1.04)		
**Stage of Disease**				
I or II	1 (ref)		1(ref)	
III	23.33	(4.86–179.29) **	22.64	(3.59–256.25) **
**Lymph Node Status**				
pN negative	1 (ref)			
pN positive	6.79	(1.73–34.82) **		
**Tumor Size (at Diagnosis) §**				
pT1 or pT2	1 (ref)			
pT3 or pT4	-	-		
**Tumor Histology**				
Ductal carcinoma	1 (ref)			
Lobular carcinoma	0.58	(0.03–6.51)		
Other	1.16	(0.04–30.63)		
**Histologic Grade**				
1	1 (ref)			
2	0.56	(0.02–16.14)		
3	1.00	(0.04–27.00)		
**Ki-67**				
<20%	1 (ref)			
≥20%	0.94	(0.27–3.30)		
**ER**				
Negative	1 (ref)			
Positive	0.97	(0.27–3.62)		
**PgR**				
Negative	1 (ref)			
Positive	0.96	(0.29–3.17)		
**Adjuvant Chemotherapy §**				
Anthracyclines and taxanes	1 (ref)			
Taxanes only	5.33	(0.08–46.61)		
Anthracyclines only	0.76	(0.10–4.32)		
Other (endocrine therapy, CMF, vinorelbine)	-	-		
**BMI at start of Adjuvant Chemotherapy**	1.21	(1.03–1.45) *	1.25	(0.99–1.65)
**HDAC6 ¥**	16.49	(2.58–164.10) **	32.78	(2.65–973.27) *
**HDAC6 †**	2.55	(1.37–5.47)	3.20	(1.38–9.91)
**WNT4 ¥**	0.93	(0.55–1.54)		
**BMPR2 ¥**	7.38	(1.19–64.35) *		
**PALB2 ¥**	0.72	(0.15–3.16)		
**PARP1 ¥**	2.55	(0.73–10.20)	0.157	

OR: odds ratio; CI: confidence interval; ER: estrogen receptor; PgR: progesterone receptor; BMI: body mass index; CMF: Cyclophosphamide, Methotrexate, Fluorouracil. ** *p*-value < 0.01; * *p*-value < 0.05. § Due to the presence of zero cells, the ORs were not computed. ¥ Reported as log2 transformed variable. † Reported as log2 transformed variable and multiplied by 3. The associated odds ratio then is for a 1/3-unit increase in log2 transformed HDAC6. The *p*-value is the same as log2 transformed only variable.

## Data Availability

The data that support the findings of this study are available from the corresponding author upon reasonable request.

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
