# Peer review of "HDAC6 as a Prognostic Factor and Druggable Target in HER2-Positive Breast Cancer"

_cancers, 2024, doi:10.3390/cancers16223752_

Round 1
Reviewer 1 Report
Comments and Suggestions for Authors
The manuscript is well written and demonstrates a novel relationship between HDAC6 and HER2 positive breast cancer. The findings are relatively novel, interesting and potentially of significant clinical value. The English usage is fine with only a few syntax errors to correct. The methodology used is scientifically sound with appropriate controls used in the main. The conclusions derived are logical and again appear sound for the most part.
There are some minor issues to address.
There would be some concerns around the fact that the study is retrospective and the numbers are relatively small. A power calculation would be helpful to reassure the reader as to the validity of the findings.
Can the authors please clarify if the patients were a symptomatic population, screening population or mix of both.
The authors do not state the exact Her2 status for each case. I understand that they are over expressing Her2 but were they all Her2 3+ scores or were they a mix of Her2 2+ and amplified by FISH.
The authors mention that based on the PAM50 subtypes the breakdown of the subtypes were 37% luminal B, 19.6% luminal A and 8.7% basal type. They acknowledge that there are reported discrepancies between immunohistochemistry and the PAM50 subtypes but this deserves some further context and a further description as to why. It would be interesting to see if the discrepant cases were all confirmed Her2 positive by FISH or DDISH. Were any Her2 mutant as well as over expressing?
The authors should be consistent with the style of their bar charts, some are two dimensional black and white, others are three dimensional and coloured.
Figure 1 Panel D is illegible due to the crowded text, please amend.
Samples had at least 50% tumour, what was the range of tumour nuclear content for the cases.
The tumour inflammation signature is mentioned but should be described further in the context of the inflammatory gene signatures seen.
Why was OS only significantly affected by HSP90AA1 and not by HDAC6.
Finally, the authors provide a good rational and pathway analysis for HDAC6 and the synergistic link with combined HDAC6 and Her2 therapies. However, a more detailed hypothesis around the role played by HDAC6 and HSP90 would be welcome, perhaps through HER2 kinase activity as touched on in the discussion?
Reviewer 2 Report
Comments and Suggestions for Authors
The work about "HDAC6 as a prognostic factor and druggable target in HER2-positive breast cancer" is interesting but there are following minor concerns that need to be addressed before publication.
1. The introduction provides a comprehensive background on HER2+ breast cancer and the challenges associated with trastuzumab resistance. However, it would benefit from a brief mention of emerging alternative therapies or novel approaches being researched beyond anti-HER2 agents to provide a broader context for the current therapeutic landscape.
2. The section discussing the molecular mechanisms of resistance to anti-HER2 therapies is well- described. However, including a figure or diagram summarizing these pathways (e.g., PI3K/AKT/mTOR, c-MET overexpression, PTEN loss) could enhance clarity and provide a visual aid for readers to better understand these complex interactions.
3. The discussion provides a thorough explanation of HDAC6's role and the observed effects of Nexturastat A in combination with trastuzumab. However, the section could benefit from addressing any potential limitations of the study, such as the need for further validation in more diverse or larger patient cohorts to strengthen the generalizability of the findings.
4. While the therapeutic implications of combining trastuzumab with HDAC6 inhibitors are well articulated, it would be useful to include a brief discussion on the potential side effects or safety concerns of this combination therapy, especially since pan-HDAC inhibitors have been associated with toxicity issues.
5. The statement about the lack of established cutoff values for HDAC6 positivity and the need for better anti-HDAC6 antibodies are important. It would be beneficial to suggest potential strategies or future directions for addressing these challenges to provide a clearer path forward for clinical application.
Comments on the Quality of English LanguageCareful revision can eliminate minor mistakes.
